# Structural polymorphisms and substrate promiscuity of a ribosome-associated molecular chaperone

Chih-Ting Huang[1], Yei-Chen Lai[2], Szu-Yun Chen[1], Meng-Ru Ho[1], Yun-Wei Chiang[2], Shang-Te Danny Hsu[1,3,*]

1. Institute of Biological Chemistry, Academia Sinica, Taipei 11529, Taiwan

2. Department of Chemistry, National Tsing Hua University, Hsichu 30013, Taiwan

3. Institute of Biochemical Sciences, National Taiwan University, Taipei 106, Taiwan

*Correspondence to: Shang-Te Danny Hsu (sthsu@gate.sinica.edu.tw)*

**Abstract. Trigger factor (TF) is a highly conserved multi-domain molecular chaperone that exerts its chaperone activity at the ribosomal tunnel exit from which newly synthesized nascent chains emerge. TF also displays promiscuous substrate binding for a large number of cytosolic proteins independent of ribosome binding. We asked how TF recognizes a variety of substrates while existing in a monomer-dimer equilibrium. Paramagnetic nuclear magnetic resonance (NMR) and electron spin resonance (ESR) spectroscopy were used to show that dimeric TF displays a high degree of structural polymorphism in solution. A series of peptides has been generated to quantify their TF binding affinities in relation with their sequence compositions. The results confirmed a previous predication that TF preferentially binds to peptide fragments that are rich in aromatic and positively charged amino acids. NMR paramagnetic relaxation enhancement analysis showed that TF utilizes multiple binding sites, located in the chaperone domain and part of the prolyl trans/cis isomerization domain, to interact with these peptides. Dimerization of TF effectively sequesters most of substrate binding sites, which are expected to become accessible upon binding to the ribosome as a monomer. As TF lacks ATPase activity, which is commonly used to trigger conformational changes within molecular chaperones in action, the ribosome-binding-associated disassembly and conformational rearrangements may be the underlying regulatory mechanism of its chaperone activity.**

## 1. Introduction

Molecular chaperones are pivotal in facilitating protein folding and maintaining proteostasis *in vivo* (Hartl, 2016; Hartl and Hayer-Hartl, 2002). In prokaryotes, trigger factor (TF) is a highly conserved multi-domain molecular chaperone, consisting of a ribosome binding domain (RBD), substrate binding domain (SBD) and a prolyl peptidyl trans-cis isomerization domain (PPI) (Hesterkamp and Bukau, 1996; Hoffmann *et al.*, 2010). TF is a unique molecular chaperone in that it is the first molecular chaperone that all newly synthesized nascent polypeptide chains encounter (Hoffmann *et al.*, 2012; Kaiser *et al.*, 2006). TF binds to the ribosomal protein L23 through the RBD in a 1:1 stoichiometry at the exit of the ribosomal tunnel where newly synthesized nascent polypeptide chains emerge during translation (Ferbitz *et al.*, 2004; Lakshmipathy *et al.*, 2007; Merz *et al.*, 2008; Rutkowska *et al.*, 2008). Unlike most molecular chaperones, which display ATPase activity to confer chaperone activities, TF does not have ATPase activity. Instead, TF forms a dragon-like cradle at the ribosomal tunnel exit to sequester emerging nascent chains and continues to hold onto its substrates after being released from the ribosome until folding is complete (Ferbitz *et al.*, 2004; Hesterkamp and Bukau, 1996). Therefore, TF is considered as a holdase to delay protein (mis)folding events. TF binds to the ribosome with a dissociation constant ($K_d$) of ca. 1 μM, and free TF self-dimerizes in solution with a comparable $K_d$. Given that the cellular concentrations of the ribosome and TF are *ca.* 15 and 50 μM,

respectively, most ribosomes are likely to be occupied by one TF molecule leaving ca. 35 μM of free TF in monomer-dimer equilibrium. In the presence of ribosome-bound nascent chains, the binding affinity of TF to the ribosome can be enhanced by up to two orders of magnitudes; the off-rate of TF from the ribosome is also markedly slowed down remarkable (Kaiser *et al.*, 2006).

Although TF primarily acts on the ribosome in a co-translational manner, TF has been shown to exhibit chaperone activity in the absence of the ribosome to promiscuously facilitate folding of cytosolic proteins. The crystal structure of a thermophilic TF in complex with the ribosomal protein L9 shows that TF binds to its substrate in a 2:2 stoichiometry, demonstrating the multifaceted substrate recognition modes of TF (Martinez-Hackert and Hendrickson, 2009). While there is good NMR evidence to demonstrate that a ribosome-bound nascent chain can fold into its native conformation without the aid of TF (Cabrita *et al.*, 2016; Cabrita *et al.*, 2009; Hsu *et al.*, 2007; Waudby *et al.*, 2019), there are indeed a handful of proteins within the *E. coli* proteome that require the contributions of TF to fold correctly (Niwa *et al.*, 2012). Although the number of obligatorily substrates of TF within the *E. coli* proteome is limited, TF plays an important role in working with SecA and SecB to regulate the membrane protein secretory pathway, as nascent membrane proteins need to be correctly sorted by TF and SecA/B as they emerge from the ribosome (Buskiewicz *et al.*, 2004; Gelis *et al.*, 2007; Huang *et al.*, 2016). It therefore raises the question as to how TF recognizes specific substrates when faced with a multitude of sequence variations of the bacterial proteome during co-translational and post-translational folding. To this end, an empirical scoring function for predicting TF binding motifs has been proposed based on peptide array analyses: a putative TF binding motif should be at least eight amino acids in length and contain both aromatic (phenylalanine, tyrosine and tryptophan) and positively charged lysine or arginine residues (Patzelt *et al.*, 2001). In a landmark study, Kalodimos and co-workers demonstrated that it requires three TF molecules to bind to one fully unfolded PhoA, which contains multiple TF binding sites with low μM binding affinities (Saio *et al.*, 2014). TF exhibits multiple substrate binding sites in SBD and PPI, which are evolutionarily conserved. The additivity of substrate binding affinities in the multiple binding sites on SBD and PPI results in much higher binding affinity and specificity for long PhoA fragments with an apparent $K_d$ in the nM range, rendering the ability to determine the solution structures of TF in complex with three different PhoA fragments by solution state nuclear magnetic resonance (NMR) spectroscopy. The structural information of TF in complex with various PhoA fragments indicated that TF preferentially binds to aromatic residues as well as large hydrophobic residues. There is no indication of the preference for positively charged lysine or arginine residues as previously predicted.

In this study, we sought to evaluate the predictive power of the empirical scoring function for TF binding motifs proposed by Bukau and co-workers (Deuerling *et al.*, 2003; Patzelt *et al.*, 2001). Combining methyl NMR and electron spin resonance (ESR) spectroscopy, we confirmed the dynamic and polymorphic nature of the TF dimer. We generated a collection of fluorescein isothiocyanate (FITC) labeled peptide to validate by fluorescence polarization (FP) the predictive power of the proposed TF binding scoring function. Two TF binding peptides were subsequently spin-labeled for paramagnetic relaxation enhancement (PRE) measurements to identify multiple substrate binding sites within the SBD and PPI. Importantly, we demonstrated that dimerization of TF can sequester these binding motifs, and that the dynamic equilibrium between monomer and dimer is essential for substrate recognition. Collectively, our findings illustrated the functional importance of TF dimerization in the context of co-translational folding.

## 2. Materials and Methods

### 2.1 Purification of recombinant TF variants

The open reading frame of *E. coli* TF was obtained from the Nara *E. coli* ORF collection (http://ecoli.aistnara.ac.jp/Resource/ResourceManage.jsp) and had been subcloned into a pET-21d plasmid with a $His_6$-tag at the N-terminus. The constructs of the SBD and the RBD-truncated TF construct corresponding to residues 113-432 (hereafter

PPI+SBD) were kind gifts from Prof. H. Jane Dyson (Scripps Institute, U.S.A.). The single cysteine mutants, Arg14Cys (14C), Thr150Cys (150C), Glu326Cys (326C), and Ser376Cys (376C), (Kaiser *et al.*, 2006) were kind gifts from Prof. F. Ulrich Hartl (Max Planck Institute for Biochemistry, Germany). The plasmids of all TF variants were amplified using an *E. coli* DH5α strain (Sigma-Aldrich, U. S. A.) with appropriate antibiotics selection, and their sequences were subsequently confirmed by standard DNA sequencing (Genomics, Taipei, Taiwan).

Unlabeled, uniformly $^{15}N$ labeled, or uniformly $^{15}N/^{13}C$ labeled protein samples were expressed by growing the transformed cells in Luria-Bertani (LB) medium or M9 minimal medium containing $^{15}NH_4Cl$ (1 g/L) and $^{13}C$ D-glucose (2 g/L) for uniformly $^{15}N/^{13}C$ labeling in the presence of kanamycin or ampicillin for antibiotics selection. Selective $^{13}C$ and $^{1}H$ labeling at methyl groups of isoleucine (δ1), leucine, valine, methionine and/or Ala β positions – U-[$^{15}N$,$^{2}H$], Ile-[δ1-$^{13}C_m$,$^{1}H_m$], [Leu/Val-[$^{13}C_m$,$^{1}H_m$], Met-[$^{13}C$,$^{1}H$] and/or Ala-[β-$^{13}C_m$,$^{1}H_m$] – was achieved by growing *E. coli* culture in perdeuterated M9
medium containing 99.9% $D_2O$, $^{15}NH_4Cl$ (1 g/L) and $^{2}H$ D-glucose (2 g/L) followed by addition of selectively $^{13}C$ and $^{1}H$ labeled metabolic precursors, and 100 mg/L $^{13}C$-labeled methionine (Cambridge Isotope Laboratory, U. S. A.) 30 min prior to IPTG induction as described previously. For selective methyl group-labeled samples, protein over-expression was carried out at 37 ºC for four hours after the addition of IPTG. For the other samples, the overexpression was induced by the addition of 0.5 mM IPTG when the cell density reached $OD_{600}$ of 0.6–0.8 followed by overnight growth at 16 ºC.

The cells were harvested by centrifugation using a Beckmann J20XP centrifuge with a JLA 8.1K rotor for 30 min with 6000 rpm at 4˚C and resuspended in buffer containing 50 mM potassium phosphate (pH 8.0) and 300 mM NaCl. The harvested cells were disrupted using a sonicator, and the cell debris and supernatant were separated by a second centrifugation step at 45,000×g for 30 min at 4 °C. The supernatant was loaded onto a prepacked 5 ml His-Trap HP column (GE Healthcare Life Science) followed by extensive wash using buffer containing 20 mM imidazole to remove protein impurities and prevent
non-specific binding. Target fusion protein was eluted using 250 mM imidazole with the same buffer background. The eluted fractions were pooled and subject to size-exclusion chromatography (SEC; HiLoad 26/60 Superdex 75, GE Healthcare Life Sciences) with 20 mM sodium phosphate (pH 7.4) and 100 mM NaCl to remove impurities to yield a purity of higher than 95 % based on visual inspection of the Coomassie Brilliant Blue-stained sodium dodecyl sulfate polyacrylamide gel (SDS-PAGE). The protein solution was aliquoted, flash-frozen by liquid nitrogen and stored at -80 ºC until further use. Unless otherwise
specified, the SEC buffer was used for all the biophysical characterizations described herein.

**2.2 Fluorescence polarization analysis of FITC-labeled peptides**

Five peptides corresponding to fragments of ICDH (Table 1) were synthesized in-house. A fraction of all these peptides were subsequently labeled with FITC at the N-terminus for fluorescence polarization (FP) measurements. All peptides (with and without FITC labeling) were purified by high performance liquid chromatography (HPLC) and validated by MALDI-TOF
mass spectrometry against their expected molecular weights. For FP analysis, FITC-labeled peptides were dissolved in DMSO to yield a stock solution of 4 M. They were subsequently diluted by 20mM Tris (pH 7.4) and 100mM NaCl to yield a molar concentration of 100 μM. 200 μl of TF variants were transferred into a 96 well plate followed by serial dilution by the same buffer in a 1:1 dilution ratio. The final protein concentrations were between 0.1 μM and 1000 μM. After serial dilution, 50 μl of protein solutions of various protein concentrations were transferred to new wells by an eight-channel pipette and were mixed
with FITC-labeled peptide solution to yield a peptide concentration of 1 μM. FP measurements of these samples were carried out using a plate reader (Paradigm, Molecular Device, U. S. A.) with an excitation wavelength of 485 nm and an emission wavelength of 535 nm. The integration time was set to 250 ms. The observed FP values as a function of protein concentration were fit to a one-site binding model using the software Prism (GraphPad, U. S. A.) to extract apparent association constants associated with different combinations of FITC-labeled peptides and TF constructs (Lou *et al.*, 2014).

**2.3 NMR paramagnetic relaxation enhancement analysis**

One mg of IcdH2 and IcdH3 peptides were individually dissolved in 1 ml deionized water and pH adjusted to 7.6. A stock solution of $S$-(1-oxyl-2,2,5,5-tetramethyl-2,5-dihydro-1H-pyrrol-3-yl)methyl methanesulfonothioate (MTSL) was prepared by dissolving MTSL powder in dimethyl sulfoxide (DMSO) to reach 150 mM. 10-fold MTSL was added to the peptide solution for overnight reaction at 4 ºC in the dark. MTSL-labeled peptides were purified by high performance liquid chromatography (HPLC) and validated by MALDI-TOF mass spectrometry against their expected molecular weights. The eluents were lyophilized and resuspended in the SEC buffer to reach a concentration of 20 mM. U-[$^{15}$N,$^{2}$H], Ile-[$\delta$1-$^{13}$C$_m$,$^{1}$H$_m$], Leu/Val-[$^{13}$C$_m$,$^{1}$H$_m$], Met-[$^{13}$C, $^{1}$H], Ala-[$\beta$-$^{13}$C$_m$,$^{1}$H$_m$] PPI+SBD and full-length TF were used for PRE measurements by recording the backbone $^{15}$N-$^{1}$H transverse relaxation optimized spectroscopy (TROSY) and side-chain methyl $^{13}$C-$^{1}$H band-selective optimized flip angle short transient heternuclear multiquantum correlation (SOFAST-HMQC) spectroscopy in oxidized and reduced states. The NMR spectra were collected by using NMR spectrometers operating at a proton Larmor frequency of 850 MHz or 600 MHz, equipped with a cryogenic triple resonance TCI probe (Bruker, Germany), processed by NMRPipe and analyzed by NMRFAM-SPARKY (https://nmrfam.wisc.edu/nmrfam-sparky-distribution/). The nitroxide of MTSL was reduced by adding an aliquot of ascorbic acid to yield a final concentration of 1 mM. The observed PREs were expressed as the ratio of the peak intensities of the oxidized (paramagnetic state) over the reduced (diamagnetic state) state ($I^{ox}/I^{red}$).

**Continuous-wave (CW) and pulsed ESR measurements**

Introduction of MTSL into single cysteine TF variants, *i.e.*, 14C, 150C, 326C and 376C, was achieved by incubating the protein samples with 10 mM DTT, which was removed by using a desalting column (PD-10, GE Healthcare, USA). 10-fold molar excess of MTSL was added immediately after the removal of DTT and the mixtures were incubated overnight at 4 ºC in the dark. Free MTSL was subsequently removed by using the same desalting column and complete MTSL incorporation was confirmed by mass spectrometry (Rezwave, Taiwan). A Bruker ELEXSYS E580-400 X-band cw/pulsed spectrometer, equipped with a split-ring resonator (EN4118X-MS3) and a helium gas flow system (4118CF and 4112HV), was used. CW ESR spectra were recorded at temperature 310 K, with an operating frequency of 9.4 GHz, 100-kHz field modulation and 1.5 mW incident microwave power. 0.25-0.6 mM TF variants in deuterated buffer were loaded in 3 mm (O.D.) quartz tubes. d8-glycerol was supplemented to achieve a final glycerol concentration of 30 % (v/v). The total volume is approximately 20 μL. For the electron spin echo (ESE) measurements, sample tube was plunge-cooled in liquid nitrogen and then transferred into the ESR probe head, which was precooled to 50 K. ESE experiments were performed using the 2-pulse Hahn echo sequence, consisting of a $\pi$/2 pulse along the x-axis followed by a delay $\tau$ and a train of $\pi$ pulses, separated by inter-pulse delays 2$\tau$ (Lai *et al.*, 2013; Zecevic *et al.*, 1998). The field was adjusted to optimize the spin echo, and the duration times of $\pi$/2 and $\pi$ pulses were set to 16 and 32 ns. As previously described [2], the ESE signals were fitted to a stretched exponential function to extract $T_2$ values from the ESE data using the MATLAB software.

For DEER measurements, samples were prepared either by single-labeled TF variants or 1:1 mixture of two different single-labeled TF variants with the final protein concentration of 0.25 mM. 30 % (v/v) d8-glycerol was added to the sample as cryoprotectants in all DEER measurements. DEER experiments were performed using the typical four-pulse constant-time DEER sequence (Jeschke, 2012). The detection pulses were set to 32 and 16 ns for $\pi$ and $\pi$/2 pulses, respectively, and the pump frequency was set to approximately 65 MHz lower than the detection pulse frequency. The pulse amplitudes were chosen to optimize the refocused echo. The $\pi$/2-pulse was employed with +x/−x phase cycles to eliminate receiver offsets. The duration of the pumping pulse was 32 ns, and its frequency was coupled into the microwave bridge by a commercially available setup from Bruker. All pulses were amplified via a pulsed traveling wave tube (TWT) amplifier (E580-1030). The field was adjusted such that the pump pulse is applied to the maximum of the nitroxide spectrum, where it selects the central $m_I = 0$ transition 10 $A_{zz}$ together with the $m_I = \pm1$ transitions. The accumulation time for each set of data was about 10 h at a

temperature of 50 K. Determination of inter-spin distance distribution of the DEER spectroscopy was performed using home-written program operating in the Matlab (MathWorks) as previously described and demonstrated (Lai *et al.*, 2019; Sung *et al.*, 2015; Tsai *et al.*, 2015). Basically, the data were analyzed using the Tikhonov regularization based on the L-curve method (Chiang *et al.*, 2005b), followed by a data refinement process using the maximum entropy method (MEM) to obtain the non-negative distance distributions (Chiang *et al.*, 2005b; Li *et al.*, 2020).

## 3.  Results

TF exists in solution in a monomer-dimer equilibrium with a low $\mu M$ $K_d$ (Kaiser *et al.*, 2006). The RBD is responsible for the dimerization (Patzelt *et al.*, 2002). Isolated RBD, SBD and PPI exhibit well-resolved 2D $^{15}N$-$^1H$ backbone amide and $^{13}C$-$^1H$ side-chain methyl correlation spectra whose chemical shifts assignments have been previously reported at a residue-specific level ; Huang and Hsu, 2016; Yao *et al.*, 2008). These assignments serve as the basis to complete the backbone and side-chain methyl NMR chemical shift assignments through a divide-and-conquer assignment strategy despite the apparent high-molecular weight of full-length TF of approximately 100 kDa (Morgado *et al.*, 2017; Saio *et al.*, 2014; Saio *et al.*, 2018). Detailed structural and dynamic analysis of full-length TF by solution state NMR spectroscopy remains very challenging not least because of the large dynamics range of the peak intensities corresponding to different domains of TF. This can be exemplified by the very large dynamic range of the cross-peak intensities of the alanine methyl resonances of full-length TF under a perdeuterated background (Fig. 1a). The methyl resonances corresponding to the alanine residues within the RBD were severely broadened whereas those of PPI remained very sharp, and those of the SBD were intermediate. The truncation of the RBD significantly reduced the dynamic range of the observed alanine methyl resonances for both the SBD and PPI+SBD (Supplement Fig. S1).

To further probe the dynamics of individual domains in the context of a dimeric TF, we employed ESR spectroscopy with site-specific spin labels. We individually introduced a spin label to one of the four sites in TF, namely residue 14 on the RBD (14C), residue 150 on PPI (150C), and residues 326 or 376 on the SBD (326C or 376C), by covalently attaching a MTSL to the mutated cysteine side-chain (Fig. 1a, inset). The spin-labeled TF variants were analyzed by CW-ESR to probe the domain dynamics manifested in the line shapes. Comparison of the CW-ESR spectra the TF variants showed distinct side bands for 14C at 310 K, suggesting the presence of multiple conformations (Fig. 1b). In contrast, the CW-ESR spectrum of 326C showed minor signals that were similar to those of 14C, while 376C did not exhibit the same signals, implying that the conformational heterogeneity of the RBD is more pronounced than that of the SBD. In the case of 150C, there was no indication of conformational heterogeneity as the ESR lines were sharp without side band. ESE analysis was subsequently used to deduce the transverse relaxation time ($T_2$) of the free radical, *i.e.*, the nitroxide of MTSL, at individual sites (Fig. 1c). In line with the methyl NMR line width analysis, the $T_2$ of 14C was the shortest (2350 ns), followed by 376C (2800 ns), 326C (2900 ns), and that of 150C was the longest (3300 ns). Further comparison of the time domain spin-echo ESR spectra of 326C and 376C in $H_2O$ and $D_2O$ showed a clear impact of solvent on the relaxation of the spin labels. The results indicated that both spin labels were solvent exposed despite their implication in dimer formation. Collectively, the NMR and ESR analyses suggested distinct domain dynamics of a dimeric TF with PPI being the least restricted, and the RBD being the most heterogeneous. Although the SBD also forms part of the dimer interface, its dynamics is less restricted than that of the RBD.

The severely broadened methyl proton resonances of the RBD residues and faster $T_2$ relaxation of the spin label at 14C likely correspond to the conformational heterogeneity within the dimer interface. Indeed, a number of different TF dimer structures have been reported by two independent studies based on different NMR restrains (Morgado *et al.,* 2017, Saio *et al.*, 2018). To investigate the TF dimer conformations through ESR spectroscopy, we carried out double electron-electron resonance (DEER) measurements to determine the inter-spin distance distributions of different combinations of spin-labeled TF samples. These included the uniformly single species or the 1:1 mixture of two variants (denoted as site A′/site B). Figure

shows the distance distributions extracted from the DEER time-domain data (Supplement Fig. S2) using the Tikhonov-based regulation methods (Lai *et al.*, 2019; Chiang *et al.*, 2005a). The DEER distance distributions (solid lines in Fig. 2) are compared with the predicted inter-spin distance distribution (shaded areas in Fig. 2) calculated from the three previously reported NMR structures (Morgado *et al.,* 2017; Saio *et al.*, 2018) using the MtsslWizard program (Hagelueken *et al.*, 2015). In general, the DEER distance distributions show multiple distinct populations indicating conformational heterogeneity in the TF-dimer. While the majority of the DEER-derived peak distributions could find correspondences from the NMR structures, a few discrepancies did exist. They were indicated by asterisks in Fig. 2. Specifically, the DEER measurements identified a shorter distance pair for 14'/14 centered at approximately 3 nm, when all reported NMR structures showed corresponding distances

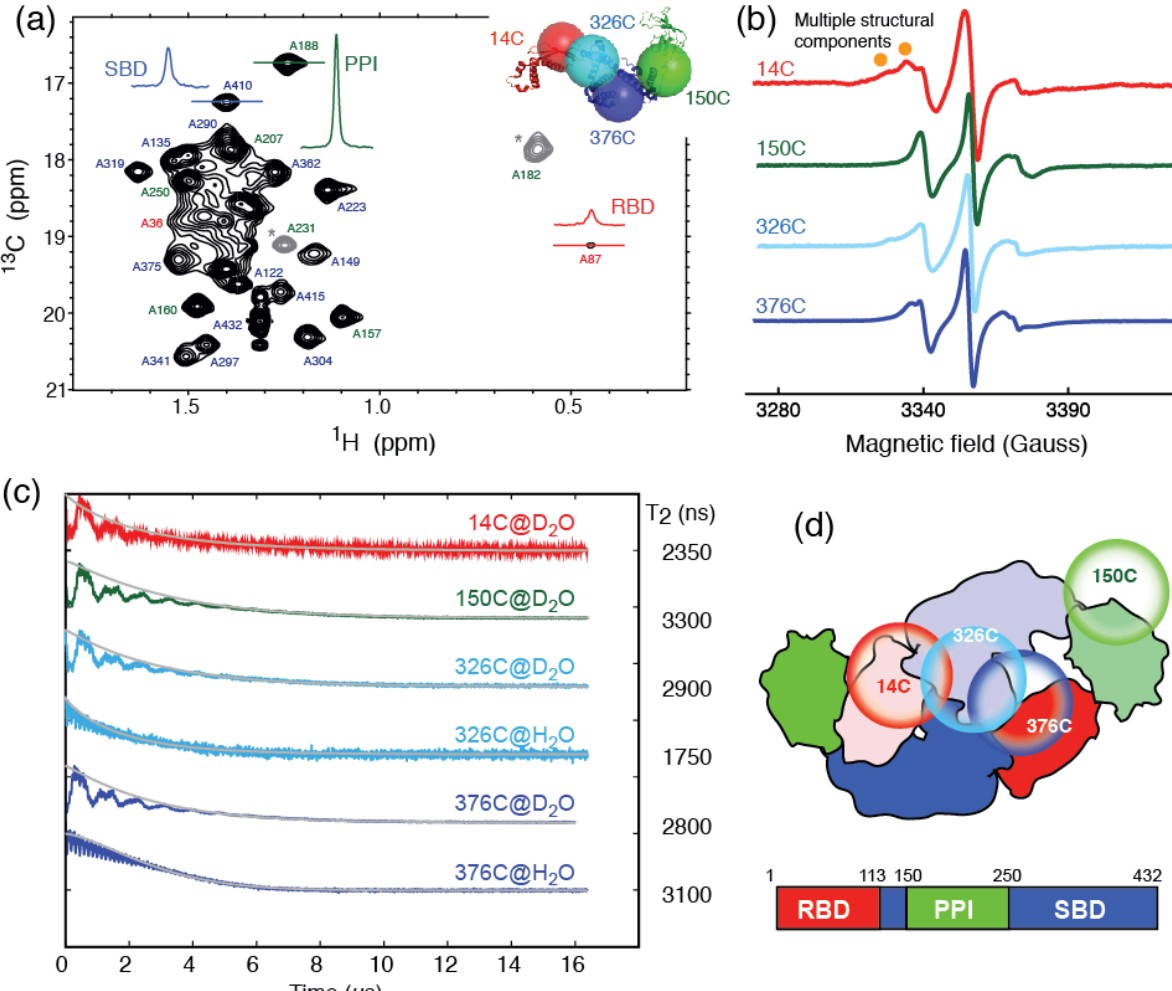

Figure 1. Domain dynamics of dimeric TF in solution. (a) $^{13}$C-$^{1}$H heteronuclear multi-quantum correlation (HMQC) spectrum of U-[$^{15}$N,$^{2}$H], Ala-[$\beta$-$^{1}$H$_{m}$/$^{13}$C$_{m}$] TF. Selected proton line shapes of cross-peaks originated from the RBD, SBD and PPI are shown in red, blue and green, respectively, with their corresponding residue identities indicated. Aliased cross-peaks are shown in grey and labeled with asterisks. Inset: Cartoon representation of a monomeric TF (PDB ID: 1w26) with matching coloring for the individual domains. The C$\alpha$ atoms of residues 14, 150, 327 and 376, which were individually mutated into cysteine for MTSL spin labeling, are shown in semitransparent spheres with a radius of 25 Å and indicated with their residue identities. (b) CW-ESR spectra of spin-labeled TF variants recorded at 310 K as a function of magnetic field. The spectrum of 14C exhibits multiple side peaks (indicated by filled orange circles) that are indicative of multiple structural components. (c) ESE measurements of spin-labeled TF variants. The transverse relaxation times (T$_{2}$) of the nitroxide were deduced from the fitting shown in solid gray lines, and their values are indicated on the right. Comparison of the relaxation characteristics of the same samples in H$_{2}$O and D$_{2}$O for 326C and 376C indicated the solvent-exposed nature of the spin labels. (d) Schematic representation of the domain arrangement of dimeric TF with the locations of the spin labels indicated in open circles. The domain organization of TF is shown below with the boundaries of individual domains indicated.

at 4 nm and above. Likewise, the DEER-derived distance distribution of 326'/326 showed an additional peak at approximately 4.7 nm, but it was not present in the NMR structures. Furthermore, the DEER-derived distance distributions of 14'/326 and 14'/376 showed three distinct populations, which were in agreement with the conclusion drawn by the CW-ESR analysis that the RBD exhibits abundant structural heterogeneity. Overall, the RBD (14C) exhibited a higher level of conformational heterogeneity than what was previously determined by NMR spectroscopy. Collectively, our ESR analyses clearly

demonstrated the abundant structural polymorphism of the TF dimer in solution.

      Having established the ground work of characterizing the dynamics of TF in its apo form, we next set to characterize how TF recognizes its substrates. According to the peptide array study based on the sequence of isocitrate dehydrogenase (ICDH), several surface-immobilized peptides showed prominent TF binding (Deuerling *et al.*, 2003). Together with the results derived from other peptide arrays, an empirical scoring function for predicting the potential TF binding site along a given

protein sequence was proposed (Deuerling *et al.*, 2003; Patzelt *et al.*, 2001). Nevertheless, the predictive power of such a scoring function has not been experimentally verified thus far. According to the prediction, an ideal TF binding motif should be at least eight residues long, and rich in aromatic residues and positively charged lysine or arginine. The requirement for the coexistence of hydrophobic and charged residues is an intriguing feature. Nevertheless, the relatively loose definition can lead to a huge number of potential binding sites within the bacterial proteomes. As a model system, we correlated the previously

reported peptide array data of TF binding to ICDH (Deuerling *et al.*, 2003) and the predicated TF binding score as a function of ICDH sequence (Fig. 3a). By visual inspection of the blotting densities of the peptide array, we identified five segments

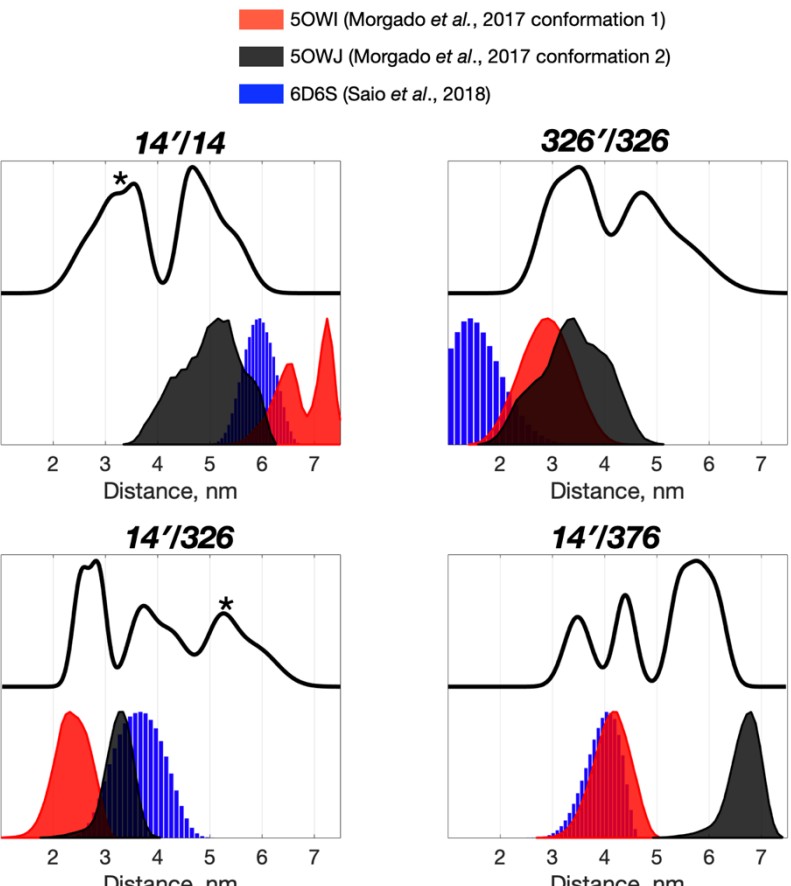

**Figure 2. Multiple dimeric TF conformations revealed from the DEER measurements**. DEER samples were prepared by either the single species or the 1:1 mixture (denoted as site A'/site B) of the three single-cysteine variants, 14C, 326C and 376C. DEER distance distributions of TF dimer (solid line) were compared with the distance distributions calculated from the previously determined TF dimer structure (PDB codes: 5OWI (red), 5OWJ (black), and 6D6S (blue)). There are a few discrepancies between the DEER and NMR results, as indicated by asterisks.

within the ICDH sequence that showed strong TF binding and fulfilled the requirement of peptide length and composition (Table 1). We adjusted the window sizes of the selected sequences to maximize the amount of preferred amino acid types and chemically synthesized these peptides followed by introducing a FITC moiety at the N-termini individual peptides to facilitate fluorescence polarization (FP) measurements to determine the binding affinities of these peptides to TF. Except for IcdH3 and IcdH4 whose sequences are partly helical in the crystal structure (Bolduc *et al.*, 1995), all the remaining sequences correspond to loop regions that do not adopt particular secondary structures. The resulting dissociation constants ($K_d$) ranged between low μM to low mM, spanning more than two orders of magnitudes (Fig. 3b). Importantly, the natural logarithms of the observed $K_d$ values showed a good correlation with the predicted binding score (an $R^2$ value of 0.88 was obtained from the linear regression), demonstrating the predictive power of the empirical scoring function (Fig. 3c).

To further examine the structural basis of substrate recognition by TF, we chose IcdH2 and IcdH3 (Table 1), which had an endogenous cysteine residue within their sequences that can be spin-labeled with MTSL for paramagnetic relaxation enhancement (PRE) measurements. We first used U-[$^{15}$N,$^2$H], Ile-[δ1-$^{13}$C$_m$,$^1$H$_m$], [Leu/Val-[$^{13}$C$_m$,$^1$H$_m$], Ala-[β-$^{13}$C$_m$,$^1$H$_m$], Met-[$^{13}$C,$^1$H] PPI+SBD, which is monomeric, to collected 2D $^{15}$N-$^1$H backbone amide and $^{13}$C-$^1$H side-chain methyl correlation spectra in the presence of the MTSL-labeled IcdH2 or IcdH3 under oxidized (paramagnetic) and reduced (diamagnetic) states to determine the PREs originated from the interaction with MTSL-labeled IcdH2 or IcdH3 defined by the resonance intensity ratios between the oxidized and reduced states, $I^{ox}/I^{red}$ (Supplement Fig. S3). The observed PREs were mapped onto the structure of PPI+SBD, which revealed multiple hotspots within the SBD and one cluster within PPI that showed strong PREs (Fig. 4a-c, Fig. 5a-c). Although the binding affinity of IcdH3 to full-length TF ($K_d$ = 132±9 μM) is weaker than that of IcdH2 ($K_d$ = 8.6±0.3 μM), the observed PREs in IcdH3 were more prominent than that of IcdH2 when PPI+SBD, which is a truncated and monomeric form of TF, was used in the NMR PRE analysis (Fig. 4 and Fig. 5). When full-length TF was used for the

**Table 1. List of ICDH-derived peptides and their properties associated with TF binding**

| Peptide name | Sequence[#] | Length (residue number) | Predicted score | Length-normalized score | $K_d$ (μM) |
|---|---|---|---|---|---|
| IcdH1 | $^{55}$KAYKGERKISWMEIYT$^{70}$ | 16 | -8.9 | -0.56 | 28.2±1.0 |
| IcdH2 | $^{125}$YICLRPVRYYQGT$^{137}$ | 13 | -14.0 | -1.08 | 8.6±0.3 |
| IcdH3 | $^{177}$KFLREEMGVKKIRFPEHC$^{194}$ | 18 | -9.9 | -0.55 | 132±9 |
| IcdH4 | $^{230}$KGNIMKFTEGAFK$^{242}$ | 13 | -5.6 | -0.40 | 293±49 |
| IcdH5 | $^{340}$GTAPKYAGQDK$^{350}$ | 11 | -1.6 | -0.13 | 1466±2117 |

# The starting and ending residue numbers corresponding to the sequence of ICDH of individual peptides are indicated as superscripts.

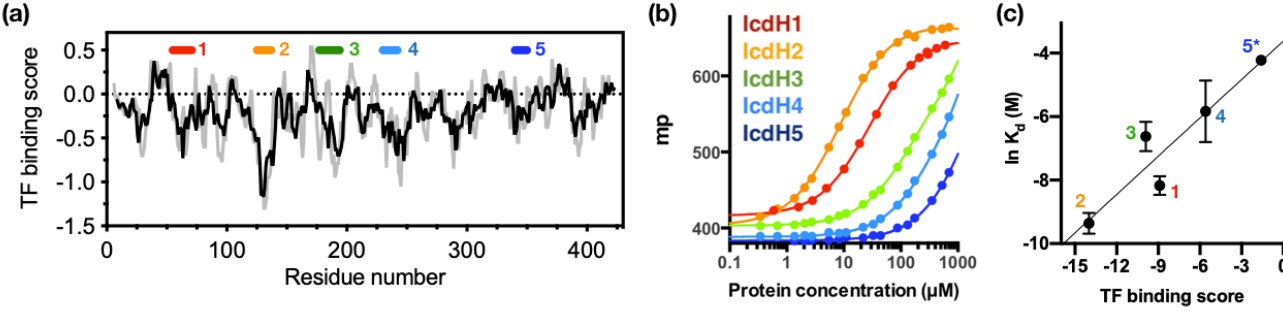

**Figure 3. Experimental validation of the scoring function of predicted TF binding motifs**. (a) Predicted TF binding score of as a function of residue number of ICDH. Running averages with a window size of eight and 13 residues are shown in grey and black, respectively. A segment with a predicted binding score of lower than -0.5 is considered as a potential binding site. Five peptides were chemically synthesized corresponding to the regions indicated above, numbered from 1 to 5. (b) Fluorescence polarization (FP) analysis of TF binding to the synthetic peptides labeled with FITC. The FP of the peptides IcdH1-5 as a function of TF concentration are colored from red to blue as indicated in inset, which correspond to the segments indicated in (a). (c) Linear regression of the natural logarithm of the binding constant ($K_d$) as a function of the predicted TF binding score. The resulting function is y = 0.0856 x + 0.772 with $R^2$ = 0.88. The error bars were derived from three technical replicates of the FP analysis. The error of IcdH5 was larger than the mean value, which is omitted in this plot

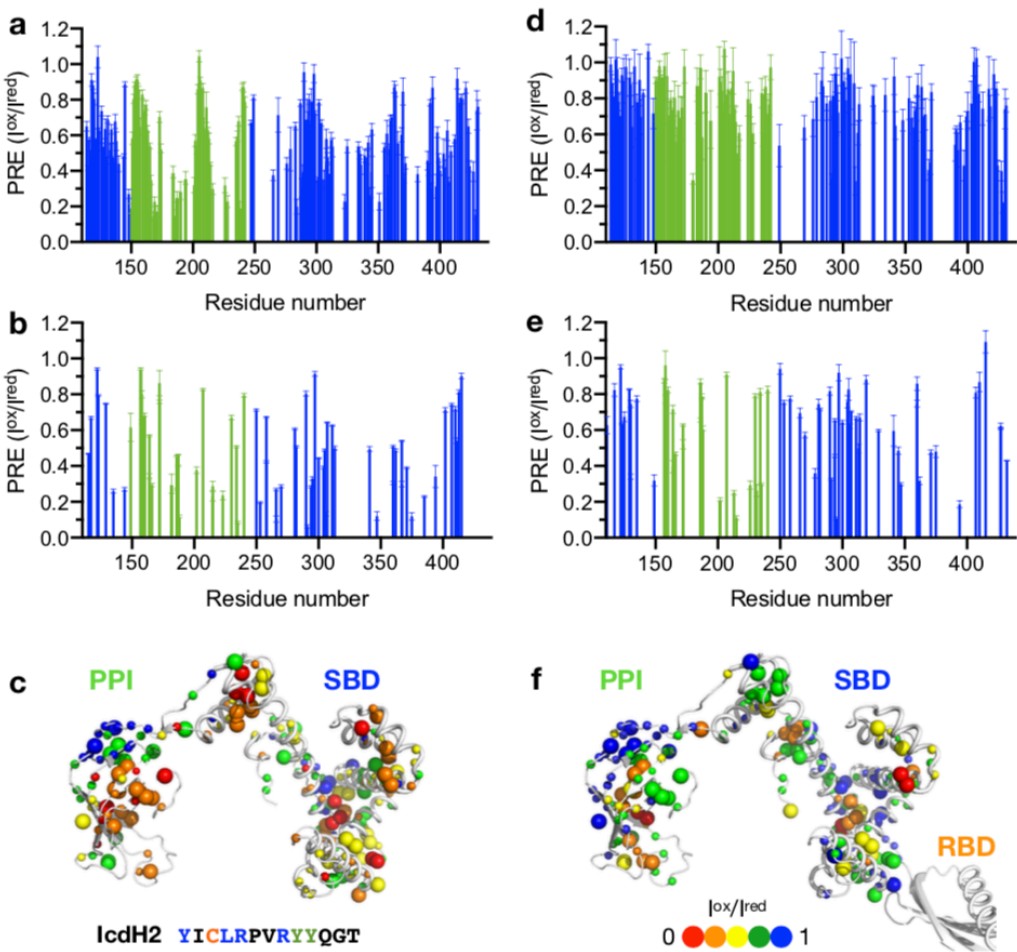

Figure 4. Structural mapping of the PREs induced by the MTSL-labeled IcdH2 peptide on TF without and with the RBD. The backbone amide-based PREs of PPI+SBD (a) and full-length TF (d). The side-chain methyl-based PREs of PPI+SBD (b) and full-length TF (e). Structural mapping of the observed PREs onto the structure of PPI+SBD (c) and full-length TF (f). The backbone amide nitrogen atoms and side-chain methyl carbon atoms are shown in small and large spheres, and are color-ramped from red to blue, corresponding to small and larger PREs as indicated by the filled circles below. The observed PREs expressed as the ratio of the peak intensities of the oxidized (paramagnetic state) over the reduced (diamagnetic state) states ($I^{ox}/I^{red}$) as a function of residue number between 113 and 432. The PRE values corresponding to PPI and SBD are colored in green and blue, respectively. The residues corresponding to the RBD are omitted due to the severe line broadening that precludes reliable data analysis.

same NMR PRE analysis under the TF concentration that it is predominantly dimeric, the PREs were significantly reduced (Fig. 4d-e, Fig. 5d-e), and the remaining PREs were mostly localized within PPI that is not part of the TF dimer interface (Fig. 4f and Fig. 5f). The loss of PRE was much more pronounced for IcdH3 compared to that of IcdH2, in line with the FP analysis that showed a weaker TF binding for IcdH3 compared to IcdH2. The implication of this finding is that the dimerization of TF sequesters the substrate binding sites within the SBD and to a lesser extent the binding site in PPI. Dynamic equilibrium between the monomeric and dimeric TF is therefore expected to play an important role in regulating its chaperone activity.

## 4. Discussion

In this study, we employed methyl NMR and ESR spectroscopy to characterize the dynamics of full-length TF in its dimeric form. Although TF exists in equilibrium between monomer and dimer, the experimental conditions under which the NMR and ESR experiments were conducted, *i.e.*, protein concentrations well above 0.25 mM, ensured that TF is predominantly dimeric, as has been established previously (Morgado *et al.*, 2017; Saio *et al.*, 2018). Our findings indicated that the TF dimer interface

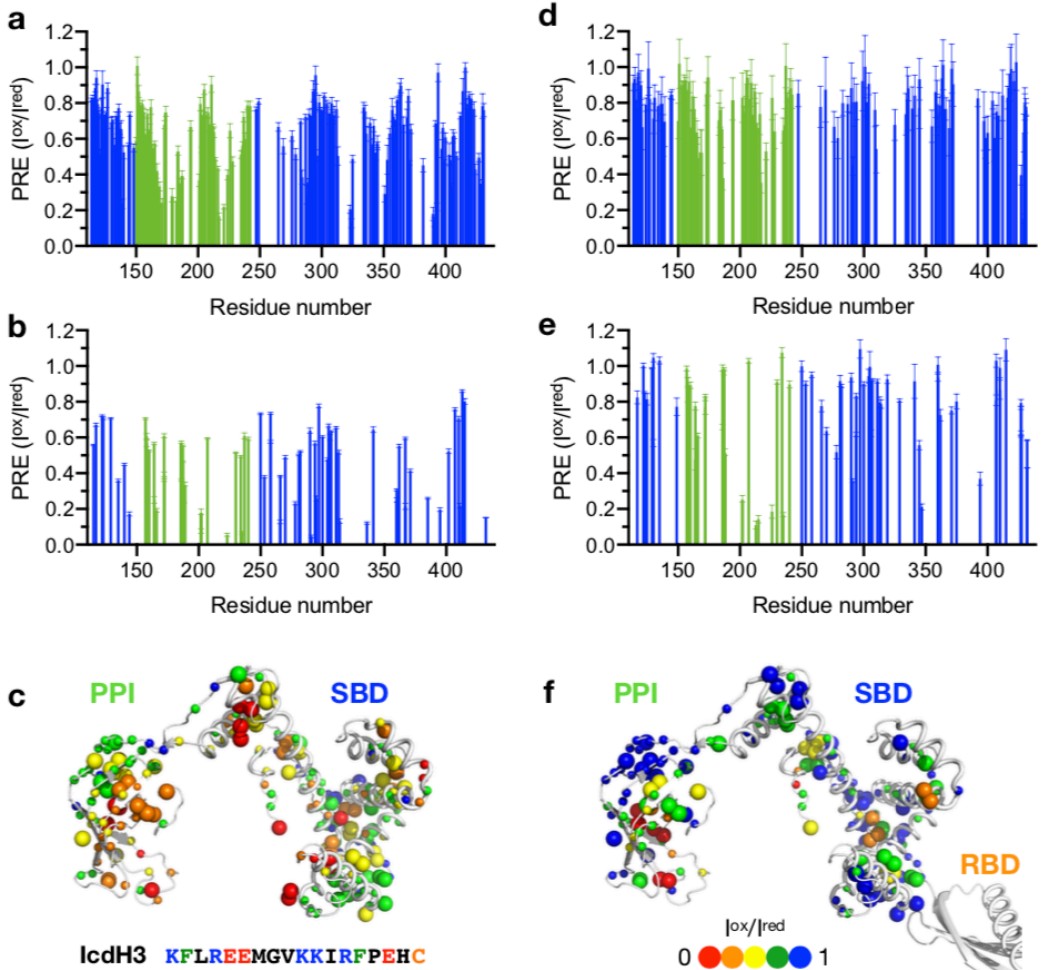

Figure 5. Structural mapping of the PREs induced by MTSL-labeled IcdH2 peptide on TF without and with the RBD. The backbone amide-based PREs of PPI+SBD (a) and full-length TF (d). The side-chain methyl-based PREs of PPI+SBD (b) and full-length TF (e). Structural mapping of the observed PREs onto the structure of PPI+SBD (c) and full-length TF (f). Structural mapping of the PRE of MTSL-labeled IcdH3 peptide on TF variants. The color scheme of the bar charts and the cartoon representations of the structural models are the same as that of Fig. 4.

is not well-defined, which may exist in several distinct configurations, as evidenced by the severely broadened linewidths of the alanine methyl group within the RBD (Fig. 1a). The conformational heterogeneity within the RBD also manifested in the additional side bands of the CW-ESR spectra of 14C, which probe the environment around the MTSL moiety at the RBD (Fig. 1b). Furthermore, there is a good correlation between the NMR methyl linewidth analysis and the T2 analysis of the MTSL-labeled TF variants, which showed that the PPI is the most dynamic, consistent with the previous findings that TF forms an antiparallel dimeric assembly where PPI makes limited contacts with other domains (Morgado *et al.*, 2017; Saio *et al.*, 2018).

We next collected DEER data on singly MTSL-labeled TFs or 1:1 mixture of TFs that were separately MTSL-labeled to measure long-range inter-spin distances up to 7 nm (Fig. 3). Unlike the nuclear Overhauser effect (NOE) or PRE that provide averaged distance information that is heavily weighted by shorter contacts, ESR-based DEER data can be converted into distance distributions of inter-spin distances, thereby informing us on the degree of conformational heterogeneity. We compared the distance distributions derived from the DEER measurements and the simulated distances based on the reported NMR structures, and showed that the NMR structures only reflect part of the conformations observed by DEER measurements (Fig. 3). Morgado *et al*. determined two distinct dimeric assemblies of TF based on PRE-derived distances (Morgado *et al.,* 2017) whereas Saio *et al*. used a large number of NOEs to determine a single ensemble of TF dimer (Saio *et al.,* 2018). Except for the intermolecular distance between 14C and 376C (14'/376 in Fig. 2) where two of the three NMR-derived TF dimers

show identical inter-spin distances, which also agreed with the DEER measurements, essentially all three reported NMR structures probe distinct subsets of conformations within a large conformational space that was probed by ESR-based DEER measurements. Nevertheless, there were notable discrepancies between the NOE-derived NMR structure and DEER measurements. On the one hand, the NOE-based NMR structure yielded a very short theoretical inter-spin distance distribution of singly labeled 326C centered at 1 nm, whereas the DEER measurement showed two major peaks centered at 3.5 and 4.7 nm, respectively (326'/326 in Fig. 2). The 3.5 nm peak distribution was in agreement with the two distinct PRE-derived TF conformations. On the other hand, each of the two PRE-based NMR structures yielded a very long inter-spin distance distribution for singly labeled 14C and the 1:1 mixture of 14C and 376C (14'/14 and 14'/376 in Fig. 2) that were not probed by the DEER measurements. Collectively, our ESR analyses underscored the structural polymorphism of the TF dimer, and the similarity and difference between the reported NMR structures themselves and in relation with the DEER-derived structural information.

We next generated five FITC-labeled peptides derived from ICDH to demonstrate the predicted power of the empirical scoring function for TF binding based on the sequence composition (Table 1 and Fig. 3). Two peptides that harbor an endogenous cysteine within the sequences, namely IcdH2 and IcdH3, were spin-labeled with MTSL to map their binding sites on TF by PRE measurements. We identified three distinct binding sites within the SBD and one binding site within the PPI (Fig. 4 and Fig. 5). The locations of these binding sites are consistent with the previous study in which four disordered fragments of PhoA are used to map the binding sites on TF (Saio *et al.*, 2014). The authors also reported multiple binding sites within the PPI and SBD when short peptides were used to map the binding sites by chemical shift perturbations and intermolecular NOEs. When a longer peptide fragment of PhoA is used as a substrate, each of the substrate binding sites within the PPI and SBD is occupied by a specific TF binding motif thereby leading to a unique binding mode that enables structure determination of the substrate-bound TF. By determining the microscopic $K_d$ values for individual binding sites, which fall within the low µM range, the authors demonstrate by relaxation dispersion analysis that the multivalency of substrate recognition significantly increases the binding affinity to a nM range. Note that in the previous study, the RBD-truncated TF variant, PPI+SBD, was used to determine the solution structures of TF in complex with different PhoA fragments based on intermolecular NOEs, while full-length TF was used to demonstrate that full length PhoA in its unfolded form can be occupied by multiple TF molecules by the attenuation of peak intensities of PhoA.

According to the ESR analysis, the spin labels within the RBD and SBD are mostly solvent exposed (Fig. 1C). Furthermore, the dimer interface appeared to be quite heterogeneous and dynamic, according to methyl NMR and CW-ESR line shape analyses (Fig. 1) and the more robust DEER measurements (Fig. 2). The unique domain architecture of TF suggests that the dimer interface does not form a properly encapsulated cavity to accommodate its substrates. Additionally, the distributions of sparsely negatively charged surfaces surrounded by small patches of neutral (hydrophobic) surfaces within the SBD and PPI coincide with the observed peptide binding sites, may explain why positively charged residues and aromatic residues are both favored for TF binding. Unlike GroEL/GroES, which has an efficient nucleotide-dependent regulatory mechanism to mechanically control the exposure of its substrate binding sites, TF may utilize the self-dimerization to achieve the same regulation (Hartl and Hayer-Hartl, 2002).

Here we compared the peptide binding induced PREs in PPI+SBD and full-length TF, and showed that dimerization of TF effectively sequesters the binding sites within the SBD from peptide binding. Although PPI is not involved in dimer formation, the peptide induced PREs in the PPI are also diminished potentially due to steric hindrance. Considering that the effective peptide binding affinities are relatively weak compared to the dissociation constant of TF self-association, it is not surprising that the TF dimerization can outcompete peptide binding at a relatively high TF concentration (100 µM). Nevertheless, cytosolic TF concentration is estimated to be in the range of 35 µM while that of the ribosome is about 1 µM. TF binds to the ribosome in a 1:1 stoichiometry, and the associated binding affinity is strongly modulated by the presence and compositions of fledgling nascent chains.

## 5. Conclusion

The intricate interplay between TF, nascent chains, and the ribosome can be modulated by the sequence compositions of the nascent chains. Our NMR, ESR and biophysical analyses confirmed the structural polymorphism of the TF dimer, and the multivalency of substrate binding, which is sensitive to TF dimer formation. These results led us to propose that the relatively strong ribosome binding affinity serves as the key regulatory mechanism to modulate monomer-dimer equilibrium and therefore the accessibility of the substrate binding sites, which are fully exposed when TF binds to the ribosome through its RBD. The observed binding affinities of the selected peptides from ICDH indeed fit well within the dynamic range of these binding events.

## Data availability

Data are available upon request

## Supplement

## Author contributions

STDH conceived and designed the experiments with contributions from YWC for ESR. CTH prepared the NMR and ESR samples and assisted NMR data collection and analysis. YCL and YWC collected and analyzed the ESR data. CTH and SYC contributed to the FITC-labeled peptide binding analyses supported by MRH. STDH wrote the manuscript with inputs from all authors.

## Competing interest

The authors declare that they have no conflict of interest

## Acknowledgements

This article is dedicated to the 80th birthday of Prof. Robert Kaptein as part of the special issue "Robert Kaptein Festschrift". We thank Prof. Ulrich Hartl at Max Planck Institute of Biochemistry, Martinsried, Germany, for stimulating discussions and sharing the constructs of the single cysteine TF variants. We also thank Prof. H. Jane Dyson at Scripps Institute, San Diego, U.S.A, for sharing the constructs of the SBD and PPI+SBD. The NMR data were collected at the High Field NMR Center at Academia Sinica, which is funded by Academia Sinica Core Facility and Innovative Instrument Project (AS-CFII-108-112), the Instrumentation Center at the National Tsing Hua University (NTHU), supported by the Ministry of Science and Technology, Taiwan (MOST) and NTHU, and the Instrument Center at the National Taiwan University. The ESR data were collected at the Instrumentation Center at NTHU. The FITC-labeled peptides were synthesized by and analyzed at the Synthesis Facility and Biophysics Facility, respectively, at the Institute of Biological Chemistry, Academia Sinica.

## Financial support

This research has been supported by MOST, Taiwan (grant numbers 100-2113-M-001-031-MY2 and 102-2113-M-001-017 -MY2 to STDH) and the intramural fund from Academia Sinica, Taiwan, to STDH.

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
