# Peer review of "Structural polymorphism and substrate promiscuity of a ribosome-associated molecular chaperone"

_Magnetic Resonance, 2021_

## Author Comment (AC1)

In the present manuscripts the authors aim at characterising the chaperone trigger factor (TF) regarding the dimerisation of TF and its implication on interaction with substrates. If the topic is interesting and trying to answer biologically relevant questions, the overall quality of the paper appears poor and too preliminary for publication.

1. There is a lengthy discussion about TF assignments. As presented here, RDB and PPI domains were assigned previously by the authors, SBD and TF113-432 by Dyson's group and the full length by Kalodimos and Hiller groups. It thus seems to me that all this constructs were assigned before. Please clarify this section and state explicitly your contribution to the field.

   As per the reviewer's suggestion, we have substantially shortened the literature review of the previous NMR work, but still acknowledge their contributions that enable us to carry out the NMR PRE analyses as described in this work.

2. For characterising the dimer interface and dynamics, the authors use a combination of NMR line-widths analysis and EPR. This is an interesting approach, however the obtained data remain low resolution and do not allow for a clear description of the dimer properties. This section is thus not very conclusive and as the authors have all the tools in hand to label with MTSL each domain of TF, I am wondering why they do not prepare mixture of isotopically labelled TF with each of the MTSL constructs successively to answer which domains of TF are in fact interacting or in close proximity. Extra EPR measurements could be used to derive informations about the domain even if more distant than the PRE distance range.

   We appreciated the reviewer's suggestion, and have now included the DEER measurements that indeed provide additional structural information that is not accessible by NMR-based analysis. The ESR-based DEER results showed that the previously reported TF dimer structures reported by Kalodimos and Hiller's groups sample part of the conformational space that is probed by ESR. The description of the DEER analysis is included in lines 196-215.

3. In the section regarding interaction with peptides, the authors aims at verifying the quality of a theoretical model model regarding TF-target interactions. This is an interesting question however the current approach lack rigorous testing. In fact the selection of the 5 constructs remains unclear to me. For exemple why is the site around 290 not chosen? It seems from the figure to be a potentially better site than 5. Also no testing on a site which is predicted to not or poorly interact is done.

   The five peptide sequences were selected not only based on the predicted binding scores but also based on the previously reported peptide array data by Deuerling *et al.*, 2003. Visual inspection of the blotting intensities of individual peptide fragments showed that the site around residue 290 has little TF binding. So we did not choose this region for peptide synthesis. To clarify this issue, we included a statement in line 224 that reads:

   *"As a model system, we correlated the previously reported peptide array data of TF binding to ICDH (Deuerling et al., 2003) and the predicated TF binding score as a function of ICDH sequence (Fig. 3a). By visual inspection of the blotting densities of the peptide array, we identified five segments within the ICDH sequence that showed strong TF binding and fulfilled the requirement of peptide length and composition (Table 1). We adjusted the window sizes of the selected sequences to maximize the amount of preferred amino acid types and chemically synthesized these peptides."*

   We hope the reviewer will find this explanation acceptable.

4. In the comparison of the interaction between IcdH2 and IcdH3, in the actual format it is almost impossible to assess the quality and relevance of the data and analysis. The PRE figures are extremely hard to read. Please adjust those figures, possibly with a multiple panel organisation so that the data readable and easily comparable between the different considered systems/probes. Please also indicate where each domain is in the sequence. For exemple, when the authors indicate that "the loss of PRE was much more pronounced for IcdH3 compared to that of IcdH2", I couldn't find any quantitative comparison or direct comparison of the experimental data. The

conclusion of this section regarding TF dimerisation seems to me quite speculative regarding the current data. It might be possible however to better reach those conclusions if the data where more adequately presented.

We appreciate the reviewer's critic of our data presentation. The PRE data are replotted individually for the backbone amide and side-chain methyl groups in Figures 4 and 5.

5. In the cross-linking section, the authors aims at distinguishing the effect of TF dimerisation on substrate recognition. In Fig 6, the gel used to control the cross-linking data show a band at 50kDa for TF in absence of BS3, while in the presence of BS3 the band is a blurry broad band at very high molecular weight (>130kDa and much higher). This indicate that the form obtained by cross-linking is not a dimer but probably a set of oligomers of TF (with 3, 4 or more TF units). To test the role of dimerisation in the interaction process the cross-linking should have given a dimer and not an oligomer. To me, this render this part of the study inconclusive.

We agree with the reviewer's comment that the dimeric assembly of TF may be affected by the chemical cross-linking, but we do not have a robust method at the moment to verify the native-like conformation of the cross-linked TF. We therefore decide to remove this part of the experimental data and the corresponding discussion to avoid misinterpretation of our data.

Considering the points above, I consider it difficult to obtain strong conclusions from this study. In the current state I do not see the added value of the current study for the state of the art of the research in the field. The last figure is quite striking to me, only the (b) part seems to be a novel contribution from the group which I believe is lagging behind current structural studies of TF already existing. I would thus not recommend this paper for publication.

Additional extra minor points

1. For readability I would strongly encourage the authors to homogenise the nomenclature regarding TF constructs and domains.

We now use RBD, PPI and SBD to refer to the individual domains and their combinations.

2. Figure S3 is missing in the file I could download

We have reorganized the figures, and double checked the cross-references in the revised manuscript.

3. Figure 3 could be moved to the SI

We have moved Figure 3 to the SI as Supplement Fig. S

4. Some references are incomplete regarding, e.g. doi numbers

We have included all the doi numbers for the references.

---

## Author Comment (AC2)

In this study, the authors have measured the structural plasticity of TF domains by carrying out the ESR studies with spin-label placed at different domains in the protein. T2 relaxation is also measured to highlight the domain dynamics. Fluorescence polarization studies were done to measure the apparent binding constant of TF with five peptides labeled with FITC. The binding affinities were correlated with TF binding scores. PREs were measured on various TF variants bound to two peptides. Crosslinked dimer of TF was tested for multiple site binding to urea-denatured MBP to propose that dimerization leads to sequestering of binding sites available in SBD. Although the manuscript discusses an important aspect of protein folding, and the work is carried out meticulously, but it lacks an explanation of various points mentioned below and I, therefore, recommend the manuscript for major revision.

Major comments:

1. The basis for choosing the positions, where four amino acids (which ones?) are mutated to Cysteine, in TF for the spin-labeling is not mentioned in the manuscript. Also, are there any structural perturbations due to Cysteine mutations?

   The four mutation sites (R14C, T150C, E326C and S376C) have been used in a previous study (Kaiser et al. Nature (2006) 444 455-460) to investigate TF binding to the ribosome and nascent chains using site specific labeling of fluorescent dyes, which are similarly bulky as the MTSL spin label. The biological functions of the single cysteine variants were considered to be similar to that of the wild type. According to the 15N-1H and 13Cm-1Hm correlation spectra of the MTSL-labeled single cysteine TF variants in their reduced states, the MTSL labeling only introduced localized chemical shift perturbations (data not shown), indicating that the introductions of the spin labels did not introduce significantly structural perturbations.

2. Figure 2b shows FP data of TF binding to five peptides. Some of the binding curves do not go all the way to saturation. How reliable is such a fitting to estimate Kd values?

   As we mentioned in the figure caption of Figure 2, the fitting of IcdH5 was indeed not very reliable due to the lack of plateau resulting in large errors. In the case of IcdH3 and IcdH4, although the plateaus have not been reached, the nonlinear regression of the dataset yielded reasonably small error estimates as shown in Table 1 and Figure 2c. We therefore considered the fitting results to be relatively reliable.

3. What are the structures of the peptides used for binding? Is there any correlation between structure of the peptide and the binding affinity to TF? It is important to look at this aspect as it will shed light on selective recognition of client proteins by TF.

   We did not characterize the structures of the peptides spectroscopically as we did not expect well-defined secondary structures to be populated due to their short lengths. Furthermore, most of the selected sequences correspond to loop regions in the reported crystal structure. We have now included a statement when describing the choice of the peptide sequences in line 230, page 8, that reads:

   "Except for IcdH3 and IcdH4 whose sequences are partly helical in the crystal structure (Bolduc *et al.*, 1995), all the remaining sequences correspond to loop regions that do not adopt particular secondary structures."

   We hope the reviewer will find this acceptable.

4. It is mentioned in the manuscript that an ideal TF binding motif should be at least eight residues long, and rich in aromatic residues and positively charged lysine or arginine. But that combination may give rise to a gigantic number of peptide sequences. What is the basis for choosing the peptide sequences used in the study?

   To clarify how we chose the peptide sequences for this study, we have revised the description in line 224, which now reads

*"As a model system, we correlated the previously reported peptide array data of TF binding to ICDH (Deuerling et al., 2003) and the predicated TF binding score as a function of ICDH sequence (Fig. 3a). By visual inspection of the blotting densities of the peptide array, we identified five segments within the ICDH sequence that showed strong TF binding and fulfilled the requirement of peptide length and composition (Table 1). We adjusted the window sizes of the selected sequences to maximize the amount of preferred amino acid types and chemically synthesized these peptides."*

We hope the reviewer will find this description acceptable.

5. What is the structural evidence that the crosslinked-dimer of TF sequesters the binding sites within SBD and the change in Kd (with respect to Native TF) is not due to loss of binding sites due to some other perturbation in structure originated due to crosslinking?

   As commented by Reviewer #1, we cannot unambiguously determine the dimeric state of the chemically cross-linked TF. We therefore removed the data and discussion about the cross-linking experiment.

6. Authors proposed a model where a long nascent chain can be occupied by multiple TF molecules, however, this would be strongly dependent on peptide sequence as the number of favorable binding residues, the 3D structure of the peptide chain, the folding kinetics would all dictate the binding to TF and is dependent on primary sequence. No experimental data has been provided to support the claim and the statement is highly speculative.

   When discussing the simultaneous binding of multiple TF to the same unfolded polypeptide, we were referring to the work by Saio et al. who showed by NMR that multiple TF can bind to unfolded PhoA.

Minor comments:

1. A domain map for the protein (in SI or in main) would have been helpful in understanding the individual domain lengths and relative positions.

   We have now included the domain organization as a schematic drawing in Fig. 1d.

2. Line 27: Spelling mistake for 'isomerization'

   We have corrected this typo.

3. Line 47: Sentence needs to be corrected. It reads: ….corrected sorted by….

   It should be "correctly sorted by…", which is now corrected.

4. Line 112: Missing word after 'eight-channel'.

   We have added 'pipette' after 'eight-channel'.

5. No gap between units and corresponding numbers at several instances in the manuscript.

   We have carefully checked and added the missing gap between units and corresponding numbers.

---

## Author Comment (AC3)

This work studies the chaperone trigger factor (TF) by NMR and EPR spectroscopy. The study addresses two topics. (1) What is the structure of the trigger factor dimer? (2) How do certain peptides interact with TF? Both these questions are of high biological / biophysical impact and have been previously addressed by other studies, as correctly cited by the authors. While the present study is following those previous works, it has nonetheless the potential to contribute valuable information on both topics and should therefore eventually be published. It requires however major revisions.

On the one hand, I support the technical issues raised by the other two referees, without wanting to rephrase them here. These should be addressed.

On the other hand, the authors should substantially strengthen the interpretation and discussion part of their data such that additional insights can be gained. Contrasts and communalities to prior work are to be spelled out explicitly.

Regarding topic 1, the structure of the TF dimer has been studied previously by two independent studies (Morgado et al. Nat Comm 2017 and Saio et al. eLife 2018). Interestingly, while both studies identified the same global arrangements of domains, they came to opposite results regarding the dynamics of the complex. The Morgado study comes up with the finding that the dimer forms a multi-conformational complex. The Saio study finds that TF dimer is a single conformer. The data presented here can contribute to distinguish between the two scenarios. The authors should revise their manuscript to introduce this question in detail and to come up with an analysis as to which scenario is better (or completely) supported by their data.

We appreciate the reviewer's suggestion to clearly state the contrasts and communalities to the prior work, namely the two NMR studies of Morgado et al. that reported two conformations of TF dimer and Saio et al that reported anther conformation of TF. Through the newly included ESR DEER measurements, which report on the pair-wise distance distributions between two spin labels, we were able to demonstrate how the three NMR structures fit to the conformational space sampled by the DEER measurements.

The differences between the prior work and our current study were detailed from line 207 to line 215.

They were further discussed in the Discussion section: last paragraph of page 10.

Regarding point 2, it seems that the peptides bind in a multi-conformational ensemble at multiple sites, as evidenced by the PRE data in Figures 4 and 5. This finding should be discussed with regard to the functionality of the chaperone and contrasted more clearly to the study Saio et al 2014, where other peptides bind in a single conformation.

To address the reviewer's comment, Saio et al. also observed multiple substrate binding sites for shorter peptides. When a longer peptide fragment of PhoA that contain multiple TF binding motifs was used, the binding affinity was increased due to multivalency and a single set of binding mode was observed. We now elaborate this point in the Discussion section (line 290, page 11) that reads

*"The authors also reported multiple binding sites within the PPI and SBD when short peptides were used to map the binding sites by chemical shift perturbations and intermolecular NOEs. When a longer peptide fragment of PhoA is used as a substrate, each of the substrate binding sites within the PPI and SBD is occupied by a specific TF binding motif thereby leading to a unique binding mode that enables structure determination of the substrate-bound TF. By determining the microscopic $K_d$ values for individual binding sites, which fall within the low µM range, the authors demonstrate by relaxation dispersion analysis that the multivalency of substrate recognition significantly increases the binding affinity to a nM range."*

We hope this will help clarify the discussion of in relation with the prior work.

Minor point:

- The first abstract of the discussion is essentially an introduction. It should be moved to and merged with the introduction.

We thank the reviewer's suggestion. The first paragraph of the Discussion section is now merged into the second paragraph of the Introduction (line 46, page 2).

- Figure 7 is unsystematic, mixing affinities and lifetimes. Also, the present study adds no new insight to this Figure. It should be removed.

We have removed Figure 7 as suggested by the reviewer.

---

## Author Comment (AC4)

In the present manuscripts the authors aim at characterising the chaperone trigger factor (TF) regarding the dimerisation of TF and its implication on interaction with substrates. If the topic is interesting and trying to answer biologically relevant questions, the overall quality of the paper appears poor and too preliminary for publication.

1. There is a lengthy discussion about TF assignments. As presented here, RDB and PPI domains were assigned previously by the authors, SBD and TF113-432 by Dyson's group and the full length by Kalodimos and Hiller groups. It thus seems to me that all this constructs were assigned before. Please clarify this section and state explicitly your contribution to the field.

   As per the reviewer's suggestion, we have substantially shortened the literature review of the previous NMR work, but still acknowledge their contributions that enable us to carry out the NMR PRE analyses as described in this work.

2. For characterising the dimer interface and dynamics, the authors use a combination of NMR line-widths analysis and EPR. This is an interesting approach, however the obtained data remain low resolution and do not allow for a clear description of the dimer properties. This section is thus not very conclusive and as the authors have all the tools in hand to label with MTSL each domain of TF, I am wondering why they do not prepare mixture of isotopically labelled TF with each of the MTSL constructs successively to answer which domains of TF are in fact interacting or in close proximity. Extra EPR measurements could be used to derive informations about the domain even if more distant than the PRE distance range.

   We appreciated the reviewer's suggestion, and have now included the DEER measurements that indeed provide additional structural information that is not accessible by NMR-based analysis. The ESR-based DEER results showed that the previously reported TF dimer structures reported by Kalodimos and Hiller's groups sample part of the conformational space that is probed by ESR. The description of the DEER analysis is included in the revised manuscript (lines 196-215), as follows:

   The severely broadened methyl proton resonances of the RBD residues and faster $T_2$ relaxation of the spin label at 14C likely correspond to the conformational heterogeneity within the dimer interface. Indeed, a number of different TF dimer structures have been reported by two independent studies based on different NMR restrains (Morgado *et al.,* 2017, Saio *et al.,* 2018). To investigate the TF dimer conformations through ESR spectroscopy, we carried out double electron-electron resonance (DEER) measurements to determine the inter-spin distance distributions of different combinations of spin-labeled TF samples. These included the uniformly single species or the 1:1 mixture of two variants (denoted as site A'/site B). Figure 2 shows the distance distributions extracted from the DEER time-domain data (Supplement Fig. S2) using the Tikhonov-based regulation methods (Lai *et al.,* 2019; Chiang *et al.,* 2005a). The DEER distance distributions (solid lines in Fig. 2) are compared with the predicted inter-spin distance distribution (shaded areas in Fig. 2) calculated from the three previously reported NMR structures (Morgado *et al.,* 2017; Saio *et al.,* 2018) using the MtsslWizard program (Hagelueken *et al.,* 2015). In general, the DEER distance distributions show multiple distinct populations indicating conformational heterogeneity in the TF-dimer. While the majority of the DEER-derived peak distributions could find correspondences from the NMR structures, a few discrepancies did exist. They were indicated by asterisks in Fig. 2. Specifically, the DEER measurements identified a shorter distance pair for 14'/14 centered at approximately 3 nm, when all reported NMR structures showed corresponding distances at 4 nm and above. Likewise, the DEER-derived distance distribution of 326'/326 showed an additional peak at approximately 4.7 nm, but it was not present in the NMR structures. Furthermore, the DEER-derived distance distributions of 14'/326 and 14'/376 showed three distinct populations, which were in agreement with the conclusion drawn by the CW-ESR analysis that the RBD exhibits abundant structural heterogeneity. Overall, the RBD (14C) exhibited a higher level of conformational heterogeneity than what was previously determined by NMR spectroscopy. Collectively, our ESR analyses clearly demonstrated the abundant structural polymorphism of the TF dimer in solution

[Figure]

Figure 2. **Multiple dimeric TF conformations revealed from the DEER measurements**. DEER samples were prepared by either the single species or the 1:1 mixture (denoted as site A'/site B) of the three single-cysteine variants, 14C, 326C and 376C. DEER distance distributions of TF dimer (solid line) were compared with the distance distributions calculated from the previously determined TF dimer structure (PDB codes: 5OWI (red), 50WJ (black), and 6D6S (blue)). There are a few discrepancies between the DEER and NMR results, as indicated by asterisks.

3. In the section regarding interaction with peptides, the authors aims at verifying the quality of a theoretical model model regarding TF-target interactions. This is an interesting question however the current approach lack rigorous testing. In fact the selection of the 5 constructs remains unclear to me. For exemple why is the site around 290 not chosen? It seems from the figure to be a potentially better site than 5. Also no testing on a site which is predicted to not or poorly interact is done.

The five peptide sequences were selected not only based on the predicted binding scores but also based on the previously reported peptide array data by Deuerling *et al.*, 2003. Visual inspection of the blotting intensities of individual peptide fragments showed that the site around residue 290 has little TF binding. So we did not choose this region for peptide synthesis. To clarify this issue, we included a statement in line 224 that reads:

*"As a model system, we correlated the previously reported peptide array data of TF binding to ICDH (Deuerling et al., 2003) and the predicated TF binding score as a function of ICDH sequence (Fig. 3a). By visual inspection of the blotting densities of the peptide array, we identified five segments within the ICDH sequence that showed strong TF binding and fulfilled the requirement of peptide length and composition (Table 1). We adjusted the window sizes of the selected sequences to maximize the amount of preferred amino acid types and chemically synthesized these peptides."*

We hope the reviewer will find this explanation acceptable.

4. In the comparison of the interaction between IcdH2 and IcdH3, in the actual format it is almost impossible to assess the quality and relevance of the data and analysis. The PRE figures are extremely hard to read. Please adjust those figures, possibly with a multiple panel organisation so that the data readable and easily comparable between the different considered systems/probes. Please also indicate where each domain is in the sequence. For exemple, when the authors indicate that "the loss of PRE was much more pronounced for IcdH3 compared to that of IcdH2", I couldn't find any quantitative comparison or direct comparison of the experimental data. The conclusion of this section regarding TF dimerisation seems to me quite speculative regarding the current data. It might be possible however to better reach those conclusions if the data where more adequately presented.

We appreciate the reviewer's critic of our data presentation. The PRE data are replotted individually for the backbone amide and side-chain methyl groups in Figures 4 and 5.

[Figure]

**Figure 4. Structural mapping of the PREs induced by the MTSL-labeled IcdH2 peptide on TF without and with the RBD.** The backbone amide-based PREs of PPI+SBD (a) and full-length TF (d). The side-chain methyl-based PREs of PPI+SBD (b) and full-length TF (e). Structural mapping of the observed PREs onto the structure of PPI+SBD (c) and full-length TF (f). The backbone amide nitrogen atoms and side-chain methyl carbon atoms are shown in small and large spheres, and are color-ramped from red to blue, corresponding to small and larger PREs as indicated by the filled circles below. The observed PREs expressed as the ratio of the peak intensities of the oxidized (paramagnetic state) over the reduced (diamagnetic state) states ($I^{ox}/I^{red}$) as a function of residue number between 113 and 432. The PRE values corresponding to PPI and SBD are colored in green and blue, respectively. The residues corresponding to the RBD are omitted due to the severe line broadening that precludes reliable data analysis.